# Adsorption of Bichromate and Arsenate Anions by a Sorbent Based on Bentonite Clay Modified with Polyhydroxocations of Iron and Aluminum by the “Co-Precipitation” Method

**DOI:** 10.3390/molecules29153709

**Published:** 2024-08-05

**Authors:** Bakytgul Kussainova, Gaukhar Tazhkenova, Ivan Kazarinov, Marina Burashnikova, Aisha Nurlybayeva, Gulnaziya Seitbekova, Saule Kantarbayeva, Nazgul Murzakasymova, Elvira Baibazarova, Dinara Altynbekova, Assem Shinibekova, Aidana Bazarkhankyzy

**Affiliations:** 1Department of Chemistry, Faculty of Natural Sciences, L.N. Gumilyov Eurasian National University, Astana 010000, Kazakhstan; gaukhar-1970@mail.ru; 2Department of Physical Chemistry, Saratov State University, Saratov 410000, Russia; kazarinovia@mail.ru (I.K.); burashnikova_mm@mail.ru (M.B.); 3Department of Chemistry and Chemical Technology, Faculty of Technology, M.Kh. Dulaty Taraz Regional University, Taraz 080000, Kazakhstan; gul1970naz@mail.ru (G.S.); meirhanovna@mail.ru (S.K.); naz1282@mail.ru (N.M.); evisko_87_87@mail.ru (E.B.); altynbekova.1985@inbox.ru (D.A.); aa.shinibekova@dulaty.kz (A.S.); 4Research Institute of New Chemical Technologies, L.N. Gumilyov Eurasian National University, Astana 010000, Kazakhstan; bazarkhankyzy.a@gmail.com; 5Department of General and Biological Chemistry, Astana Medical University, Beibitshilik Str., 49a, Astana 010000, Kazakhstan

**Keywords:** bentonite, sorbent, sorption of bichromate and arsenate anions, specific surface area, polyhydroxocations, co-precipitation

## Abstract

The physicochemical properties of natural bentonite and its sorbents were studied. It has been established the modification of natural bentonites using polyhydroxoxides of iron (III) (mod.1_Fe_5-c) and aluminum (III) (mod.1_Al_5-c) by the “co-precipitation” method led to changes in their chemical composition, structure, and sorption properties. It was shown that modified sorbents based on natural bentonite are finely porous (nanostructured) objects with a predominance of pores of 1.5–8.0 nm in size. The modification of bentonite with iron (III) and aluminum compounds by the “co-precipitation” method also leads to an increase in the sorption capacity of the obtained sorbents with respect to bichromate and arsenate anions. A kinetic analysis showed that, at the initial stage, the sorption process was controlled by an external diffusion factor, that is, the diffusion of the sorbent from the solution to the liquid film on the surface of the sorbent. The sorption process then began to proceed in a mixed diffusion mode when it limited both the external diffusion factor and the intra-diffusion factor (diffusion of the sorbent to the active centers through the system of pores and capillaries). To clarify the contribution of the chemical stage to the rate of adsorption of bichromate and arsenate anions by the sorbents under study, kinetic curves were processed using equations of chemical kinetics (pseudo-first-order, pseudo-second-order, and Elovich models). It was found that the adsorption of the studied anions by the modified sorbents based on natural bentonite was best described by a pseudo-second-order kinetic model. The high value of the correlation coefficient for the Elovich model (R^2^ > 0.9) allows us to conclude that there are structural disorders in the porous system of the studied sorbents, and their surfaces can be considered heterogeneous. Considering that heterogeneous processes occur on the surface of the sorbent, it is natural that all surface properties (structure, chemical composition of the surface layer, etc.) play an important role in anion adsorption.

## 1. Introduction

An acute problem at present is the pollution of the aquatic environment with heavy metals and their derivatives, which poses a great threat to health and environmental sustainability [1,2]. Chromium and arsenic, as the main ions of heavy metals, have received considerable attention worldwide because of their strong toxicity and high mobility [3,4]. According to the regulations of the World Health Organization, the safe concentration of chromium, the permissible concentration of Cr (VI) in industrial wastewater, is 0.1 ppm, and for arsenic is 0.01 mg/L [5]. Arsenic is highly toxic in its inorganic form, and its effects can cause cancer, as it is the most carcinogenic element [6,7,8], whereas hexavalent chromium is highly toxic [9,10].

The presence of these pollutants in wastewater and groundwater can be explained by both natural and anthropogenic sources, such as mining and agriculture, as well as various industries [2,11]. High concentrations of these metals endanger the quality of water resources and the health of thousands of people and can lead to cancer in the human body even at lower levels of exposure [12,13]. To solve this problem, various wastewater treatment technologies containing arsenic and chromium anions have been developed, including extraction, adsorption, and electrochemical oxidation [14,15]. Among these approaches, adsorption is the most promising strategy for removing pollutants from aquatic environments. To reduce water pollution, anions of chromium and arsenic have been fabricated and tested for various adsorbents, including activated clay [16], biochar clay [17], modified zeolites [18], nanocomposites [13], kaolinite clay [19,20,21], and active carbon [22]. Despite the fact that these traditional adsorbents demonstrate excellent anion removal efficiency, water treatment still has some disadvantages, such as low removal ability, low stability, and poor regeneration. Therefore, it is important to develop new adsorbents that contain arsenic and chromium anions for waste disposal.

Adsorption methods using synthetic polymer sorbents [23], such as magnetic sorbents based on polyvinyl alcohol, metallic iron, and polydentate phosphase-containing extractants [24], have become promising alternatives with advantages such as the absence of sediment, ease of use, and the possibility of regeneration of the adsorbent.

The high chemical activity of bentonites allows chemical modification methods to control the adsorption properties of sorbents based on them, and the plasticity and astringency of clays makes it possible to create complex granular nanostructured sorbents combining three water purification methods at once: mechanical, sorption, and ion exchange.

Bentonite clays have a unique and complex porous structure [25], which determines their remarkable chemical activity and adsorption abilities. This structure, characterized by a hierarchy of pores of different sizes [26], arises from the basic building blocks and location of clay minerals in bentonite. The complex and hierarchical porous structure of bentonite is the basis of its remarkable properties. This structure, with its large surface area, accessibility, and reactive sites, makes bentonite a versatile material for various applications, ranging from environmental remediation and catalysis to pharmaceuticals and construction. Understanding and managing this porous structure [27] is key to unlocking the full potential of bentonite through various technological advances.

Thus, modifications are needed, such as joining with materials like iron and aluminum, which change the physical and chemical structure of the adsorbents that penetrate the clay structure, thereby improving the adsorption abilities of bentonite clays.

The adsorption of chromium and arsenic anions using a sorbent based on bentonite clay modified by the precipitation of iron and aluminum by the “co-precipitation” method is a promising solution for removing these contaminants from water sources [28]. The use of bentonite clay as a sorbent in combination with the modification of iron and aluminum rods increases the adsorption capacity of the sorbent and its efficiency in removing chromium and arsenic anions. In addition, the “co-precipitation” method provides a more stable and uniform distribution of iron and aluminum rods in the bentonite clay structure, which leads to an increase in surface area and greater accessibility for the adsorption of chromium and arsenic ions. Using this modified sorbent, it is possible to effectively study the adsorption equilibrium and kinetics of chromium and arsenic ions. This can help determine the optimal conditions for achieving maximum adsorption efficiency.

The “co-precipitation” method is a modified “sol–gel” method. Both these methods belong to the group of pillarization (inter-calling) methods [29], which aim at the chemical and physical restructuring of the clay mineral structure to increase the adsorption capacity or create spaces that promote the adsorption of specific ions. The “precipitation” method, compared with the “sol-gel” method, is a more effective method of clay modification due to its ability to provide uniform and controlled introduction of a modifier, strong interfacial interaction, higher specificity, simplicity, and cost-effectiveness. These advantages make it the preferred choice for the development of modern clay-based materials with improved performances in various applications.

The purpose of this work is to develop sorbents based on bentonite modified with polyhydroxocations of iron (III) and aluminum by the “co-precipitation” method to increase their sorption capacity with respect to chromium and arsenic anions.

## 2. Results and Discussion

### 2.1. Characteization of the Adsorbent

#### Elemental Composition of the Studied Sorbents

We analyzed the chemical and mineral compositions of the modified bentonite-based sorbents. Table 1 shows the elemental compositions of the studied sorbents modified with polyhydroxocations of iron (III) and aluminum (III) by the “co-precipitation” method. A quantitative analysis of the elemental composition was performed on an energy-dispersive X-ray fluorescence spectrometer EDX-720 (SHIMADZU, Kyoto, Japan) using fundamental parameters.

The data in Table 1 confirm that the sorbents based on natural bentonite were aluminosilicates. An increase in the concentration of the modifying component led to an increase in the concentration of the corresponding element in the bentonite sample. This increase occurs as a result of the substitution of exchanged bentonite cations, particularly calcium cations. It was not possible to determine the presence of sodium and magnesium in the modified bentonite and sorbent samples. This is because of the measurement range of energy dispersion. The X-ray fluorescence spectrometer EDX-720 is in the range of Na to U; therefore, a small amount of light metals in the samples can be taken as a background by the device and therefore not diagnosed. Figure 1 shows the X-ray diffractograms of the initial bentonites and the modified sorbents based on them. An XRD analysis was performed on a DRON-8T diffractometer (Saint Petersburg, Russia) using an X-ray tube with a copper anode (Cu-Kα radiation).

As inferred from the acquired diffractograms, the supplementary incorporation of aluminum (III) and iron (III) polyhydroxocations into bentonite using the “co-precipitation” technique does not result in alteration of the mineral and phase com-position of bentonite (in all instances examined, minerals such as montmorillonite, α-cristobalite, and plagioclase are detected).

### 2.2. The Porous Structure of the Studied Sorbents

The results of the study of the porous structure of natural bentonites and sorbents based on those modified with polyhydroxocations of iron (III) and aluminum (III) by the “co-precipitation” method are presented in Table 2. It can be seen from the table data that the modification of bentonites leads to an increase in the number of micro- and mesopores and a decrease in the number of macropores compared with the original bentonites. A large proportion of the pores of all modified samples account for pores with a size of 1.5–8.0 nm.

Such pore size redistribution also led to a significant increase in the specific surface area of the modified sorbents, and the modification of bentonite with iron (III) and aluminum polyhydroxocations also led to an increase in the specific surface area of the sorbent samples to 68 and 82 m^2^/g, respectively.

### 2.3. Adsorption Studies

The samples of the studied sorbents weighing 1–2 g were filled with distilled water for 1 h, decanted, and filled with 100 mL of a model solution containing the studied anions (bichromate and arsenate anions) of various concentrations, mixed with the model solution, and kept for 2 h until an equilibrium state was reached in the solution. The samples were then collected from the middle layers of the solution. Quantitative analysis of the elemental composition of the sample was performed on an energy-dispersive X-ray fluorescence spectrometer (EDX-720) using calibration curves. The solution was neutral at pH 7. The adsorbent dose was 1–2 g.

#### 2.3.1. Study of the Sorption Kinetics

The sorption characteristics of the modified bentonite-based sorbents were analyzed. An important characteristic in the study of the adsorption process is the kinetics of adsorption, which is necessary to determine the time required to establish adsorption equilibrium when removing adsorption isotherms. An important characteristic in the study of the adsorption process is the kinetics of adsorption, which is necessary, first, to determine the time of establishment of adsorption equilibrium when removing the adsorption isotherms and second, to establish the sorption mechanism. Oxygen-containing anions were selected as test anions: bichromate and arsenate anions were selected as test anions in the study of the adsorption process by the modified sorbents based on the studied bentonites obtained by the “co-precipitation” method. The sorption experiment technique was as follows: samples of sorbents weighing 1–2 g were filled with distilled water for 1 h; then, the water was decanted and 100 mL of a model solution of potassium arsenate and potassium di-chromate of a certain concentration was poured; and the adsorbent was mixed with the model solution. Samples were collected from the middle layers of the solution for 5, 10, 15, 20, 30, 60, 120, and 180 min. A quantitative analysis of the sample for the content of bichromate and arsenate anions was performed using an energy-dispersive X-ray fluorescence spectrometer (EDX-720) using calibration curves. The data on the kinetics of the adsorption of arsenic and chromium (III) anions on the studied sorbents, obtained based on bentonite from the Pogodayevo deposit, are shown in Figure 2a,b. The analysis of the kinetic data on the sorption of bichromate and arsenate anions on the studied sorbents indicates that the saturation of sorbents with cations under these conditions occurs for 2 h. Consequently, in the future, when removing the sorption isotherms, the time required to establish adsorption equilibrium will be 2 h.

The rate of heterogeneous reactions (sorption of solutes from a solid-phase solution) is determined by several factors. Some of these are the properties of the sorbent and sorbate. There are several stages during which absorption of a substance occurs. This stage depended on the diffusion coefficient of the substance in the external solution; then comes the stage of overcoming the boundaries (the solution is a thin film of liquid on the surface of the sorbent granules, and the liquid film is the solid phase) separating the sorbent from the sorbate. The next stage is the distribution of sorbent molecules inside the solid phase, which depends on both the properties of the substance molecule (size, charge value, hydration) and the properties of the sorbent (type, number of charges per unit mass of the matrix, pore size, moisture content, etc.). The last stage is ion exchange [28].

#### 2.3.2. Study of the Sorption Mechanism

Heavy metal ion sorption is a complex process that is regulated by a number of factors. Possible processes include chemisorption, complexation, and adsorption on the sorbent surface and its pores, as well as complexation, ion exchange, microprecipitation, and precipitation of heavy metal hydroxides [30].

The study of isotherms is the primary method for investigating adsorption mechanisms. Sorption isotherms depict the distribution of metal ions at equilibrium between the adsorbent and liquid phases as a function of concentration. The investigation of these isotherms helps us reach conclusions about the nature of the sorbent surface and the sorbate–sorbent interactions. Langmuir model was used to conduct these experiment.

The adsorption isotherms of bichromate and arsenate anions on the studied sorbents were taken as follows: as in the kinetic experiments, the samples of the studied sorbents weighing 1–2 g were filled with distilled water for 1 h; then, the water was decanted and filled with 100 mL of a model solution of bichromate and arsenate anions of various concentrations (100, 200, 300, 400, and 500 mg/L); and were kept for 2 h until the equilibrium concentration in the solution was reached. Samples were collected from the middle layer of the solution. A quantitative analysis of the elemental composition of the sample was also performed using an energy-dispersive X-ray fluorescence spectrometer (EDX-720) using calibration curves.

According to the average values of equilibrium concentrations (at least two parallel measurements), the adsorption value was calculated using the following Formula (1):*A* = (*C_i_* − *C_e_*) × *V*/*m*(1)

*A*—adsorption capacity of the sorbent, mg/g;

*C_i_*—initial concentration of the studied ions in solution, mg/L;

*C_e_*—equilibrium concentration of the studied ions in solution, mg/L;

*V*—volume of the test solution, L;

*m*—the mass of the sorbent taken for analysis, g.

Figure 3a,b show the adsorption isotherms of bichromate and arsenate anions on the studied sorbents obtained on the basis of natural bentonite deposit. Ranges of different concentrations show the true picture of the surface. All the obtained isotherms belong to Langmuir-type (L-type) isotherms. The Langmuir adsorption isotherm equation, derived based on molecular kinetic theory and ideas about the monomolecular nature of the adsorption process, when applied to solutions, has the form of Equation (2):(2)$$A=\frac{A\infty \cdot K\;\cdot Cp}{\left(1+K\;\cdot Cp\right)}$$
where *K* is the adsorption equilibrium constant characterizing the adsorption energy;

*C_p_*—equilibrium concentration, mg/L;

*A_∞_*—the maximum adsorption value, mg/g.

Adsorption isotherms are a useful tool for understanding the nature of sorbent surfaces. The Langmuir adsorption isotherm (2) is linearized in coordinates 1/*A* = *f* (1/*C*), which allows for grapho-analytical determination of the values of the coefficients K and *A_∞_*. The obtained adsorption isotherms were processed in accordance with the Langmuir equation in inverse coordinates, according to Equation (3):(3)$$\frac{1}{A}=\frac{1}{A_{\infty }}+\frac{1}{A_{\infty }K}\cdot \frac{1}{C}$$

Figure 4a,b show the adsorption isotherms of bichromate and arsenate anions for the studied sorbents in inverse coordinates, in accordance with Equation (3).

Using regression equations, the values of the maximum adsorption of the studied anions on the initial bentonite and on the modified sorbents are determined, the values of which are presented in Table 3.

As follows from the data obtained, modification of bentonite with iron (III) and aluminum compounds using the “co-precipitation” method leads to a significant increase in the value of the maximum adsorption of the studied anions. It should be noted that arsenate anions exhibit the greatest sorption activity of the studied anions: the maximum adsorption value is higher on an aluminum-modified sorbent, which reaches 14.7 mg/g.

### 2.4. Kinetic Analysis of the Adsorption Processes of Bichromate and Arsenate Anions Occurring on Bentonites Modified by Polyhydroxo Complexes of Metals

The rate of a heterogeneous reaction, which involves the sorption of solutes from a solid phase solution, is influenced by numerous factors. Among these factors are the characteristics of both the sorbent material and the solutes being absorbed. The absorption process unfolds in several distinct stages, the initial of which is contingent upon the diffusion coefficient of the solute within the surrounding solution. Subsequently, there is a phase where boundaries must be traversed, wherein a thin liquid film covers the surface of the sorbent granules, separating them from the sorbent material. Following this, there is a step involving the dispersion of sorbent molecules within the solid phase, influenced by various factors, such as the size, charge, and hydration of the solute molecule, as well as characteristics of the sorbent material, like type, charge density, pore size, and moisture content. Lastly, there is a stage involving ion exchange. The complex, multistage nature of the sorption process complicates the comprehensive consideration of all phases simultaneously, leading to the employment of kinetic models to identify the limiting stage of the process. Mathematical models including external (4) and internal diffusion (5), reactions following pseudo-first- (6) and pseudo-second-orders (7), as well as the Elovich Equation (8), are utilized for determining the limiting stage of the sorption kinetics.
*F* = *q_t_*/*q_e_*(4)
*q_t_* = *K_p_t*^0.5^ + *C*,(5)
*ln*(*q_e_* − *q_t_*) = *lnq_e_* − *k*_1_*t*,(6)
*t*/*q_t_* = 1/*k*_2_*q_e_*2 + *t*/*q_e_*,(7)
*q_t_* = 1(*lnαβ*)/*β* + 1/*β lnt*,(8)
where *q_t_* is the adsorbed amount at time *t*, mg/g^−1^;

*q_e_* is the adsorbed amount in equilibrium, mg g^−1^;

*k*_1_ is the pseudo-first-order adsorption rate constant, min^−1^;

*k*_2_ is the rate of adsorption constant of the pseudo-second order, g (mg^−1^ h^−1^);

*α* is the initial adsorption rate, mg/g·min;

*β* depends on the degree of surface coating and the activation energy of chemisorption, mg/min;

*k_p_* is the rate constant of intraparticle diffusion, mg (g^−1^ h^−0.5^).

The study of sorption kinetics allows one to identify the aspects that influence the dynamics and limit the pace of the process. The sorption rate is an important feature that determines whether the examined substance can be used as a sorbent. In general, the interaction of a sorbate with a sorbent has a high rate in the early minutes of contact, and thereafter settles to a consistent level. To reduce the sorption cycle time in a technological or laboratory process, it is preferable to attain sorption equilibrium soon [28,31].

It can be observed from Figure 5 that the kinetic profiles of the adsorption of bi-chromate and arsenate anions by the investigated modified sorbents exhibit linear behavior in the –*ln*(1 − *F*) versus t coordinates. This observation suggests that the sorption mechanism of these sorbents was primarily governed by external diffusion at the onset. However, as the process advanced, the linearity of the curve diminished, indicating an escalation in the intra-diffusion component. Consequently, the progression of the process occurs through a combination of diffusion mechanisms, specifically involving diffusion in the solution film and diffusion within the sorbent grain [32,33].

The evidence indicating that the stage that restricts the sorption process is internal diffusion is demonstrated by the observation of a linear relationship in *q_t_* coordinates with *t*^0.5^ (Figure 6). The quantity of anions that have been sorbed over time in a process controlled by diffusion can be mathematically described by Equation (5) [34].

As can be seen from Figure 7, these dependencies are multilinear and do not intersect the origin. It follows from the above that in the initial period of time, the sorption process is controlled by an external diffusion factor, i.e., refers to the diffusion of the sorbent from the solution to the liquid film on the surface of the sorbent. The sorption process then begins to proceed in a mixed diffusion mode when it limits both the external diffusion factor and the internal diffusion factor (diffusion of the sorbent to the active centers through the system of pores and capillaries). The segment cut-off by the continuation of this straight line reflects the ion exchange process occurring between the functional groups of sorption materials (bentonites) and bichromate and arsenate anions.

The parameters for the adsorption of bichromate and arsenate anions by the modified sorbents obtained using kinetic diffusion models are presented in Table 4.

To clarify the contribution of the chemical stage to the rate of adsorption of bichromate and arsenate anions by the sorbents under study, kinetic curves (Figure 2) were processed using equations of chemical kinetics (models: pseudo-first-order (6), pseudo-second-order (7) and Elovich (8).

In Figure 7, Figure 8 and Figure 9, the dependencies of *ln*(*q_e_* − *q_t_*) on *t*, *t*/*q_t_* on *t*, and *q_t_* on *lnt* are presented. It can be seen that the kinetic equations of the pseudo-first-order satisfactorily (R^2^ > 0.7) describe the experimental data, and the models of the pseudo-second-order of Elovich have a linear dependence with a correlation coefficient of 0.9 and higher.

Table 5 shows the results of processing experimental kinetic adsorption curves of the studied anions with chemical kinetics models for sorbents based on the modified bentonites.

From the data presented in Table 5, it can be seen that the adsorption of the bichromate and arsenate anions by the modified sorbents based on natural bentonite is best described by a pseudo-second-order kinetic model, as evidenced by the correlation coefficient (R^2^ > 0.9), which allows us to conclude that the contribution of sorbate–sorbate interactions during sorption anions contributes to the overall rate of the process [35].

Based on the high value of the correlation coefficient for the Elovich model (R^2^ > 0.9), it can be concluded that there are structural disorders in a porous system of materials, and their surface can be considered a heterogeneous system, since the Elovich equation is used exclusively to describe the kinetics of adsorption of substances in heterogeneous systems, considering the sorption capacity. Considering that heterogeneous processes occur on the surface of the sorbent, it is natural that all the surface properties (structure, chemical composition of the surface layer, etc.) also play an important role in adsorption [33].

## 3. Materials and Methods

The bentonite clay was obtained from the Pogodayevo deposit (West Kazakhstan region, Republic of Kazakhstan) and purified by precipitation in combination with ultrasound treatment and centrifugation. The aluminum chloride (AlCl_3_·6H_2_O, 97%), iron chloride (FeCl_3_·6H_2_O, 97%), silver nitrate (AgNO_3_, 98%), and sodium hydroxide (NaOH, 98%) were purchased from Merck (Boston, MA, USA). Potassium arsenate and potassium dichromate salt (K_3_AsO_4_, K_2_Cr_2_O_7_ 98%) were used as the model solutions.

Quantitative elemental composition analysis was conducted using an energy-dispersive X-ray fluorescence spectrometer EDX-720 (SHIMADZU, Kyoto, Japan) through the methodology of its fundamental parameters. The porous structure of the specimens was assessed via low-temperature nitrogen adsorption using a high-speed gas sorption analyzer (Quantachrome NOVA, Fremont, CA, USA). The specific surface areas of the solid samples were calculated using the Brunauer (Emmett) Teller method. The pore volume and size distribution were determined using the Barrett (Joyner) Halenda method. Initial data for calculations employing the BJH method were derived from the desorption or adsorption branch of the isotherm within the pressure range of 0.967 to 0.4 P/Po. The X-ray phase analysis was performed using an X-ray diffractometer DRON-8T (Saint Petersburg, Russia). The powder was ground in an agate mortar and placed in a quartz cuvette 2 mm deep. CuKa radiation, a Goebel parabolic mirror (AXO Dresden GmbH, Dresden, Germany) and a position-sensitive Mythen 2R1D detector with 640 channels (Dectris, Baden, Switzerland) with a discreteness of 2θ 0.0144° were used to register the diffractograms. Geometry of the focal beam: axial slits of 12 mm, equatorial 0.25 mm. The registration was carried out by rotating the cuvette 0.3 rpm in the range of angles 2° from 5 to 80° at points with a step for the central channel of the detector and exposure time at the point: 0.1°, 2c. The ability of the studied samples to absorb salt anions was determined by constructing sorption isotherms using variable concentrations under statistical conditions. The model solution was a K_3_AsO_4_, K_2_Cr_2_O_7_ salt solution. The arsenic and chromium anions (III) were analyzed using atomic absorption spectroscopy (AAS, SHIMADZU-6800).

### Preparation of Fe Bentonite and Al Bentonite

Modification of the bentonites was carried out by the method of “co-precipitation” (intercalation, or pillarization). FeCl_3_ and AlCl_3_ salts were added to the aqueous suspension of bentonite (the ratio of solid to the liquid phase is 1:10, and the pH of the aqueous extract of the suspension is 8). The concentration of iron (aluminum) in bentonite was 5 mmol Me^3+^/g. The suspension was then ultrasonicated at a frequency of 22 Hz for 3 min. [28]. Next, 0.5 M of NaOH solution was added to the prepared suspension ([OH^−^]/[Me^3+^] = 2.23), and the suspension was subjected to aging at room temperature during the day. After 24 h, the resulting modified bentonite was separated from the liquid phase on a Buchner funnel using a vacuum pump, washed with water until it reacted negatively with chloride ions, and dried at 80 °C. The washed samples were stored in an airtight container and labeled.

## 4. Conclusions

The physicochemical properties of natural bentonite and the sorbents were studied. It has been established that the modification of natural bentonite with polyhydroxocations of iron (III) and aluminum by the method of “co-precipitation” leads to a change in their chemical composition, structure, and sorption properties. It has been established that modified sorbents based on natural bentonite are finely porous (nanostructured) objects with a predominance of pores of 1.5–8.0 nm in size. The specific surface area of the sorbents depends on the nature of the modifying component being introduced (iron (III) or aluminum). The specific surface area of the Al-modified sorbents reached 82 m^2^/g, which significantly exceeded those of the Fe-modified sorbents and initial bentonite. It was shown that the modification of bentonite with polyhydroxo cations leads to an increase in the sorption capacity of the obtained sorbents with respect to bichromate and arsenate anions. The obtained sorption isotherms were classified as Langmuir-type. Arsenate anions exhibit the greatest sorption activity of the studied anions: the maximum adsorption value on the Al-modified sorbent reaches 14.7 mg/g, which is almost 5 times higher than the adsorption capacity of pure bentonite. Kinetic analysis showed that, at the initial stage, the sorption process was controlled by an external diffusion factor, that is, the diffusion of the sorbent from the solution to the liquid film on the surface of the sorbent. The sorption process then begins to proceed in a mixed diffusion mode when it limits both the external diffusion factor and the intra-diffusion factor (diffusion of the sorbent to the active centers through the system of pores and capillaries). To clarify the contribution of the chemical stage to the rate of adsorption of bichromate and arsenate anions by the sorbents under study, kinetic curves were processed using equations of chemical kinetics (pseudo-first-, pseudo-second-order, and Elovich models). As a result, it was found that the adsorption of the studied anions by the modified sorbents based on natural bentonite was best described by a pseudo-second-order chemical model, as evidenced by the correlation coefficient (R^2^ > 0.9), which allowed us to conclude about the contribution of sorbate–sorbate interactions during the sorption of anions to the overall rate of the process. Based on the high value of the correlation coefficient for the Elovich model (R^2^ > 0.9), it can be concluded that there are structural disorders in a porous system of materials and that their surfaces can be considered heterogeneous. Considering that heterogeneous processes occur on the surface of the sorbent, it is natural that all surface properties (structure, chemical composition of the surface layer, etc.) play an important role in the process of anion adsorption.

## Figures and Tables

**Figure 1 molecules-29-03709-f001:**
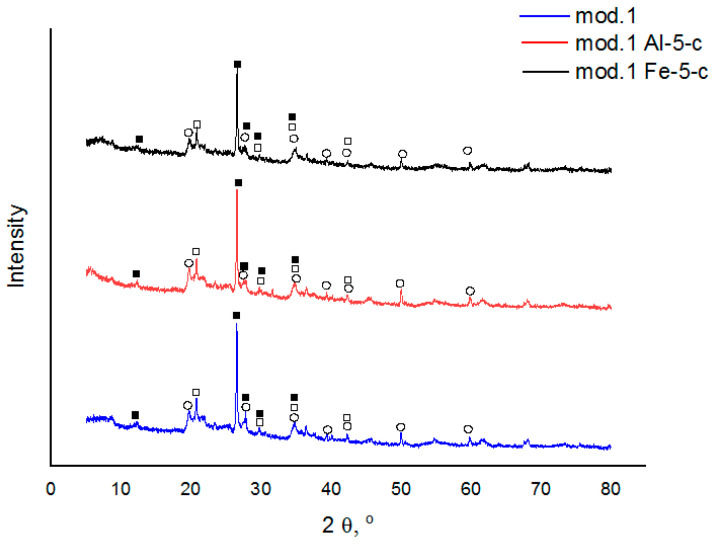
X-ray diffraction patterns of the examined sorbents derived from natural bentonite from the Pogodayevo deposit (Kazakhstan): 1—mod.1; 2—mod.1_Al_5-c; 3—mod.1_Fe_5-c, where 
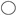
—montmorillonite; □—α-cristobalite; ■—plagioclase.

**Figure 2 molecules-29-03709-f002:**
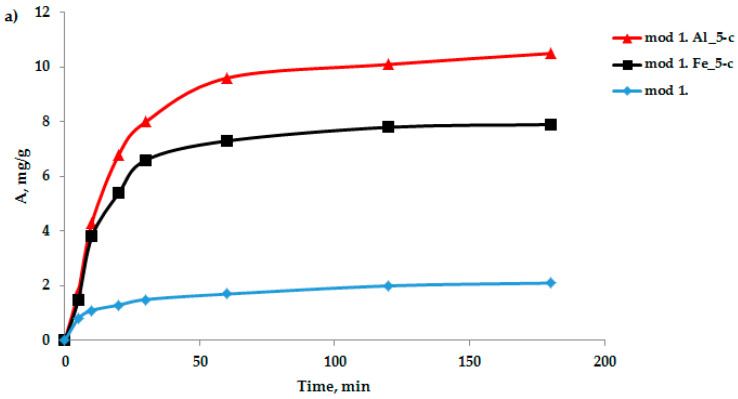
Kinetic curves of the adsorption process of bichromate (**a**) and arsenate anions (**b**) by the studied modified sorbents in a neutral medium.

**Figure 3 molecules-29-03709-f003:**
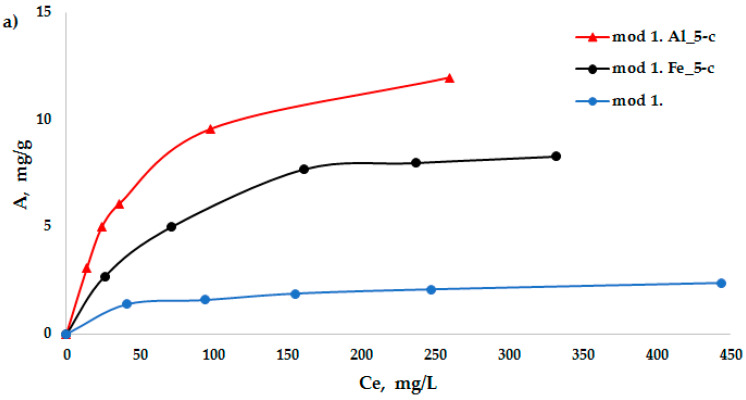
Adsorption isotherms in a neutral medium of (**a**) bichromate and (**b**) arsenate anions on the obtained sorbents.

**Figure 4 molecules-29-03709-f004:**
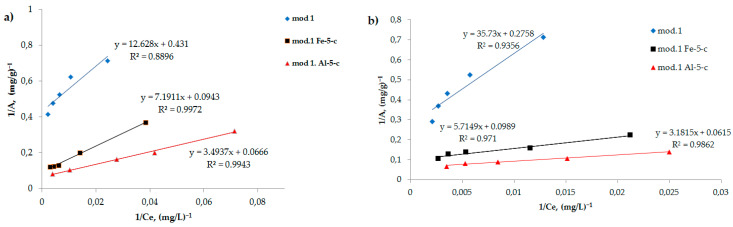
Isotherms of adsorption of (**a**) bichromate and (**b**) arsenate anions by the studied sorbents, represented in inverse coordinates in accordance with the Langmuir Equation (3).

**Figure 5 molecules-29-03709-f005:**
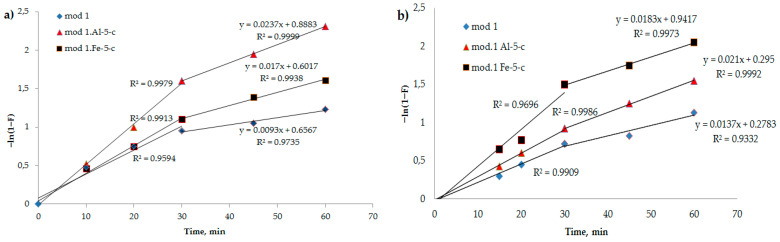
Dependence of –*ln*(1 − *F*) on t (spring diffusion model) during adsorption of bichromate and arsenate anions by bentonite and modified iron(III) and aluminum polyhydroxocations by bentonite-based sorbents: (**a**) bichromate anion; (**b**) arsenate anion.

**Figure 6 molecules-29-03709-f006:**
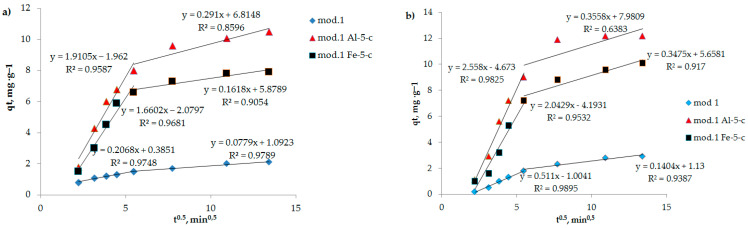
Dependence of qt on *t*^0.5^ (internal diffusion model) during adsorption of bichromate and arsenate anions by bentonite and modified iron (III) and aluminum polyhydroxocations by bentonite-based sorbents: (**a**) bichromate anion; (**b**) arsenate anion.

**Figure 7 molecules-29-03709-f007:**
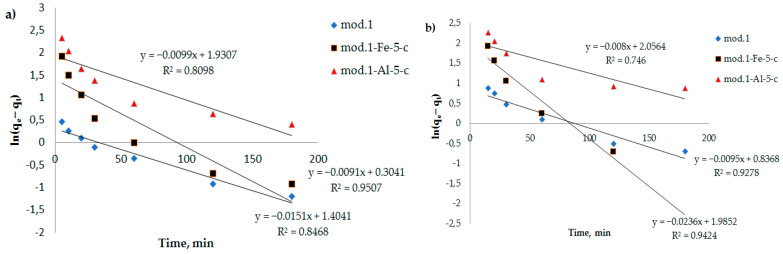
Dependence of *ln*(*q_e_* − *q_t_*) on *t* (a pseudo-first-order kinetic model) during adsorption of bichromate and arsenate anions by bentonite and modified iron (III) and aluminum polyhydroxocations by bentonite-based sorbents: (**a**) bichromate anion; (**b**) arsenate anion.

**Figure 8 molecules-29-03709-f008:**
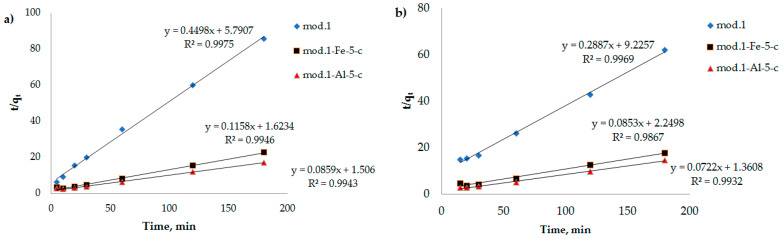
Dependence of *t*/*q_t_* on *t* (kinetic model of pseudo-second-order) during adsorption of bichromate and arsenate anions by bentonite and modified iron (III) and aluminum polyhydroxocations by bentonite-based sorbents: (**a**) bichromate anion; (**b**) arsenate anion.

**Figure 9 molecules-29-03709-f009:**
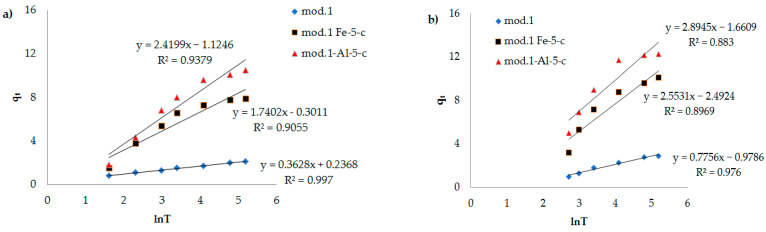
Dependence of *t*/*q_t_* on *t* (Elovich model) during adsorption of bichromate and arsenate anions by bentonite and modified iron (III) and aluminum polyhydroxocations by bentonite-based sorbents: (**a**) bichromate–anion (**b**) arsenate anion.

**Table 1 molecules-29-03709-t001:** Elemental composition of the studied bentonite-based sorbents.

Samples of Sorbents	The Content of the Element, wt. %						
	Al	Fe	Si	Ca	K	Ni	Ti
mod.1	7	47	18	4	12	10	2
mod.1_Al_5-c	19	46	14	2	9	8	2
mod.1_Fe_5-c	7	71	8	2	4	6	2

**Table 2 molecules-29-03709-t002:** The main characteristics of the porous structure of the studied sorbents based on bentonite modified with polyhydroxocations of iron (III) and aluminum (III) by the “co-precipitation” method.

Sample	Specific Surface Area, m^2^/g	Pore Volum, cm^3^/g	Distribution of Pores by Radius, %			
			1.5–2.0 nm	2.0–4.0 nm	4.0–8.0 nm	More than 8.0 nm
mod. 1	31	0.054	6	15	22	57
mod. 1_Al_5-c	82	0.083	17	37	15	33
mod. 1_Fe_5-c	68	0.071	12	26	18	44

**Table 3 molecules-29-03709-t003:** Values of the maximum adsorption capacity of the studied anions and cations for the studied bentonite-based sorbents.

Sample	*A*_∞_, mg/g	
	Arsenate Anions	Bichromate Anions
mod.1	3.4	2.4
mod.1_Al_5-c	14.7	12.0
mod.1_Fe_5-c	9.5	9.0

**Table 4 molecules-29-03709-t004:** Results of processing experimental kinetic adsorption curves of bichromate and arsenate anions with modified bentonite-based sorbents by diffusion models.

Sample	The Model of External Diffusion	Internal Diffusion Model		
	R^2^	*k_p_*	C	R^2^
arsenate anions				
mod. 1	0.9909	0.511	1.13	0.9895
mod. 1_Al_5-c	0.9986	2.558	7.981	0.9825
mod. 1_Fe_5-c	0.9696	2.043	5.658	0.9532
bichromate anions				
mod. 1	0.9594	0.2068	1.098	0.9748
mod. 1_Al_5-c	0.9979	1.9105	6.819	0.9587
mod. 1_Fe_5-c	0.9913	1.6602	5.879	0.9681

**Table 5 molecules-29-03709-t005:** Results of processing experimental kinetic adsorption curves of bichromate and arsenate anions by chemical kinetics models for sorbents based on modified bentonites.

Sample	Pseudo-First-Order			Pseudo-Second Order		
	*q_e_*	*k* _1_	R^2^	*q_e_*	*k* _2_	R^2^
arsenate anions						
mod.1	2.309	0.0095	0.9277	3.464	0.2887	0.9969
mod.1_Al_5-c	7.818	0.0080	0.7460	13.850	0.0722	0.9932
mod.1_Fe_5-c	7.281	0.0236	0.9424	11.723	0.0853	0.9867
bichromate anions						
mod.1	1.355	0.0091	0.9507	2.223	0.4498	0.9975
mod.1_Al_5-c	6.894	0.0099	0.8098	11.641	0.0859	0.9943
mod.1_Fe_5-c	4.072	0.0151	0.8468	8.635	0.1158	0.9946
**Elovich model**						
	**α**		**β**		**R^2^**	
arsenate anions						
mod.1	0.363		1.289		0.976	
mod.1_Al_5-c	0.0235		0.345		0.883	
mod.1_Fe_5-c	0.0044		0.392		0.897	
bichromate anions						
mod.1	0.395		2.756		0.997	
mod.1_Al_5-c	0.159		0.413		0.938	
mod.1_Fe_5-c	1.030		0.575		0.906	

## Data Availability

Data are contained within the article.

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
