# Peer review of "Adsorption of Bichromate and Arsenate Anions by a Sorbent Based on Bentonite Clay Modified with Polyhydroxocations of Iron and Aluminum by the “Co-Precipitation” Method"

_molecules, 2024, doi:10.3390/molecules29153709_

Round 1

Reviewer 1 Report

Comments and Suggestions for Authors

The manuscript has a practical orientation and is of interest to science, but needs some improvements.

Introduction

1. In the Introduction section there is no description of synthetic polymer sorbents, which are widely used for the sorption of heavy metals from water bodies (see works https://doi.org/10.3390/gels8080492, https://doi.org/10.1038/s41598-024 -54969-y and others). The current state of research in the field of sorbent development should be assessed in order to highlight the relevance of the chosen area of ​​research.

2. What determines the choice of bentonite as a sorbent for heavy metals in terms of sorption capacity in comparison with synthetic and biosorbents? It is known that upon contact with water, bentonite swells and forms a highly stable colloidal suspension, which is difficult to separate from the aqueous system.

Results and its discussion.

3. The mechanism of sorption of heavy metals using modified bentonite should be given.

4. Lines 189-192 and 192-195 contain the same sentences.

5. The authors provide a sorption technique, which is already in the Materials and Methods section (lines 198-205, 240-246).

6. There is no explanation for the obtained kinetic curves of the adsorption process (Figure 2). Add an explanation to the text.

7. The word sorbent should be replaced with sorbate (lines 217,218).

8. What is the reason for the higher adsorption capacity for arsenate anions compared to dichromate anions, as well as modified bentonite?

9. There is no discussion of the results in the Results and Discussion section. Please add your professional comments regarding the results obtained.

conclusions

10. It is desirable to provide prospects for further research of the sorbent for use in the field of purification of industrial wastewater from chromium and arsenic, and also to note whether it is possible to reuse the sorbent and indicate methods for recycling the spent sorbent.

Comments on the Quality of English Language

Minor editing of English language required

Reviewer 2 Report

Comments and Suggestions for Authors

This work reported a “co-precipitation” method for the modification of natural bentonites using polyhydroxoxides of iron (III) 23 (mod.1_Fe_5-c) and aluminum (III) (mod.1_Al_5-c). In general, it is a good and concise manuscript. I recommend a minor revision before the publication. 1. The advantages of "co-precipitation" method in materials modification should be introduced. 2. The crystallinity and purity of the prepared samples are suggested to be discussed from XRD characterizations.

3. The porous structure has significant role in chemical reactions and adsorption. Some recent works (Chemical Society Reviews, 2017, 46(2), 481-558., CCS Chemistry, 2021, 3(3), 2280-2297., ACS Central Science, 2023, 9(8), 1499-1503.) about porous functional materials are encouraged to be cited. 

Round 2

Reviewer 1 Report

Comments and Suggestions for Authors

The authors took an extremely responsible approach to improving the manuscript and took into account all the recommendations. I believe the manuscript can be published.